# Spectral descriptors for bulk metallic glasses based on the thermodynamics of competing crystalline phases

Eric Perim[1,2,*], Dongwoo Lee[3,*], Yanhui Liu[4,*], Cormac Toher[1,2], Pan Gong[4], Yanglin Li[4], W. Neal Simmons[1,2], Ohad Levy[1,2], Joost J. Vlassak[3], Jan Schroers[4] & Stefano Curtarolo[1,2]

Metallic glasses attract considerable interest due to their unique combination of superb properties and processability. Predicting their formation from known alloy parameters remains the major hindrance to the discovery of new systems. Here, we propose a descriptor based on the heuristics that structural and energetic 'confusion' obstructs crystalline growth, and demonstrate its validity by experiments on two well-known glass-forming alloy systems. We then develop a robust model for predicting glass formation ability based on the geometrical and energetic features of crystalline phases calculated *ab initio* in the AFLOW framework. Our findings indicate that the formation of metallic glass phases could be much more common than currently thought, with more than 17% of binary alloy systems potential glass formers. Our approach pinpoints favourable compositions and demonstrates that smart descriptors, based solely on alloy properties available in online repositories, offer the sought-after key for accelerated discovery of metallic glasses.

[1] Department of Mechanical Engineering and Materials Science, Duke University, Durham, North Carolina 27708, USA. [2] Center for Materials Genomics, Duke University, Durham, North Carolina 27708, USA. [3] School of Engineering and Applied Sciences, Harvard University, Cambridge, Massachusetts 02138, USA. [4] Department of Mechanical Engineering and Materials Science, Yale University, New Haven, Connecticut 06511, USA. * These authors contributed equally to this work. Correspondence and requests for materials should be addressed to S.C. (email: stefano@duke.edu).

Understanding and predicting the formation of multi-component bulk metallic glasses (BMGs) is crucial for fully leveraging their unique combination of superb mechanical properties[1] and plastic-like processability[2–4] for potential applications[5–8]. The process underlying the formation of BMGs is still to be fully understood. It involves a multitude of topological fluctuations competing during solidification across many length scales[9–12]. Long-range processes, required by the typical non-polymorphic nature of the crystallization, and atomic-scale fluctuations, precursors of short-range ordered competing phases[13], are all pitted against each other and against glass formation[9,14,15]. Simulations of amorphous phases have been attempted to disentangle the mechanism of glass formation[16–23], within reasonable system sizes, using classical and semi-empirical potentials. Although they have been successful in investigating the influence of factors such as the atomic size and packing on the glass-forming ability (GFA), questions about competing crystalline phases and the dynamics of the process still remain, especially considering all the approximations demanded for performing long molecular dynamics simulations. Further-more, adopting *ab initio* methods has been challenging[24]: even while the most relevant metastable crystalline phases can be calculated and sorted by their energies[25–29], the zero-temperature formalism, lacking vibrational free energy[30], and the absence of an underlying lattice on which to build configurational thermodynamics[31,32] make the problem impervious to direct computational analysis.

Descriptors for bulk glass formation—correlations between the outcome (glass formation) and other material properties, possibly simpler to characterize[24]—have been proposed based on structural[21,22,33,34], thermodynamic[8,34–38], kinetic[36,39] and electronic structure considerations[21,40]. A few of these[8,38] have been considerably successful in correlating with the GFA. However, they rely on experimental data, such as the (reduced) glass transition temperatures, that can only be obtained once the glass has been synthesized, and, therefore, cannot be used to make predictions for systems that have not yet been experimentally studied. Consequently, a definite and clear picture for predicting GFA still remains to be found.

In a seminal paper[41], Greer speculated that 'confusion' during crystallization promotes glass formation. However, challenges in *a priori* knowledge and ability to quantify such confusion have left this direction mostly unexplored. In this work we propose a definition of this 'confusion' based on the following consideration. During quenching, crystal growth will occur whenever fluctuations lead to the formation of a crystalline nucleus larger than a critical size. Therefore, to obtain an amorphous solid, the formation of critical size nuclei has to be hampered. We postulate that the existence of multiple phases with very similar energy, implying similar probabilities of being formed, but dissimilar structures, will lead to the formation of several distinct clusters, which will intimately compete and thus keep each other from reaching the critical size needed for crystallization. To demonstrate the power of this ansatz, we first characterize confusion by the approximate thermodynamic density of distinct structural phases of metastable states, obtained from *ab initio* calculations (Fig. 1), and concurrent GFA measurements by combinatorial synthesis of alloy libraries and high-throughput nanocalorimetry. As test systems, we focus on CuZr and NiZr. Among the known BMGs, CuZr is probably the most broadly studied[42–45]. NiZr, on the other hand, is known for having poor GFA[46,47]. The contrast between the two glass formers, one strong and one weak, corroborates our ansatz. After having established the efficacy of our approach, we extend it into a robust numerical model for building GFA spectra. This extension establishes the strength of our approach, leading to a descriptor that requires no experimental input and is computationally predictable, inexpensive and quick to calculate.

## Results

**Using databases for materials discovery.** Carrying out electronic structure *ab initio* calculations for the infinite number of available states for a given alloy system is obviously impossible, especially when no lattice model can be built[31,32], as in the case of BMGs. Therefore, we adopt the agnostic approach of exploring structural prototypes mostly observed in nature for these types of systems. The method, shown to be capable of reasonably sampling the phase space and predicting novel compounds[25–28,32], is expected to estimate the thermodynamic density of states of an alloy system. We use the binary alloy data available in the AFLOW set of repositories[48,49] to count the number of different structural phases in a given formation enthalpy range as a function of the composition. These data were obtained utilizing the VASP[50–52] code within the AFLOW computational materials design framework[30,53], at the density functional theory level of approximation. The binary alloy systems are fully relaxed in accordance with the AFLOW standard settings[53], which uses the GGA-PBE[54,55] exchange correlation, PAW potentials[56,57], at least 6,000 **k**-points per reciprocal atom and a plane wave cutoff at least 1.4 times the largest value recommended for the VASP potentials of the constituents. The multiple different crystalline phases for each particular stoichiometry are built from the AFLOW library of common prototypes[30].

**A simple descriptor for glass formation.** To quantify the level of disorder associated with an alloy system, we identify the most stable structures and count all of the available phases at the corresponding compositions, ordered by their formation enthalpy difference above the respective ground state, $\Delta H$. This leads to a cumulative distribution of the number of phases, $N_P(\Delta H)$ (Fig. 2). We also count the number of different Bravais lattice types $N_{BL}(\Delta H)$ and space groups $N_{SG}(\Delta H)$ among the phases in the distribution. These three quantities are combined into a single heuristic descriptor, called the 'entropic factor', $\chi_F(\Delta H)$, defined as the cubic root of their product:

$$\chi_F(\Delta H) = \sqrt[3]{N_P(\Delta H) \times N_{BL}(\Delta H) \times N_{SG}(\Delta H)} \qquad (1)$$

$\chi_F(\Delta H)$ should be related to the configurational entropy at a given composition but, by taking into account the different symmetries available to the system, it is more generally representative of the frustration of the crystallization of a single homogeneous crystal structure. Compositions with large $\chi_F(\Delta H)$ are expected to present structures with more disorder, thus leading to high GFA. In this analysis, the formation enthalpies, Bravais lattices and space groups were determined from the calculated energies and symmetries of the relaxed relevant structures.

X-ray diffraction and scanning electron microscopy measurements were performed on ingots of CuZr and NiZr alloys prepared by arc-melting the pure elements under an argon atmosphere. The alloys were re-melted and suction cast into a wedge-shaped cavity in a copper mould. The as-cast rods were cut into half along the longitudinal direction and polished to a mirror finish followed by etching. GFA was evaluated by observing the contrast change along the longitudinal direction under a scanning electron microscope. The critical thickness was determined at the transition from featureless contrast to a clearly observable microstructure, as shown in Fig. 2. The crystalline and amorphous structures were further identified by X-ray diffraction using a Cu-$K_\alpha$ source.

We also synthesized and characterized thin-film samples deposited by magnetron-sputtering elementary targets (99.99%

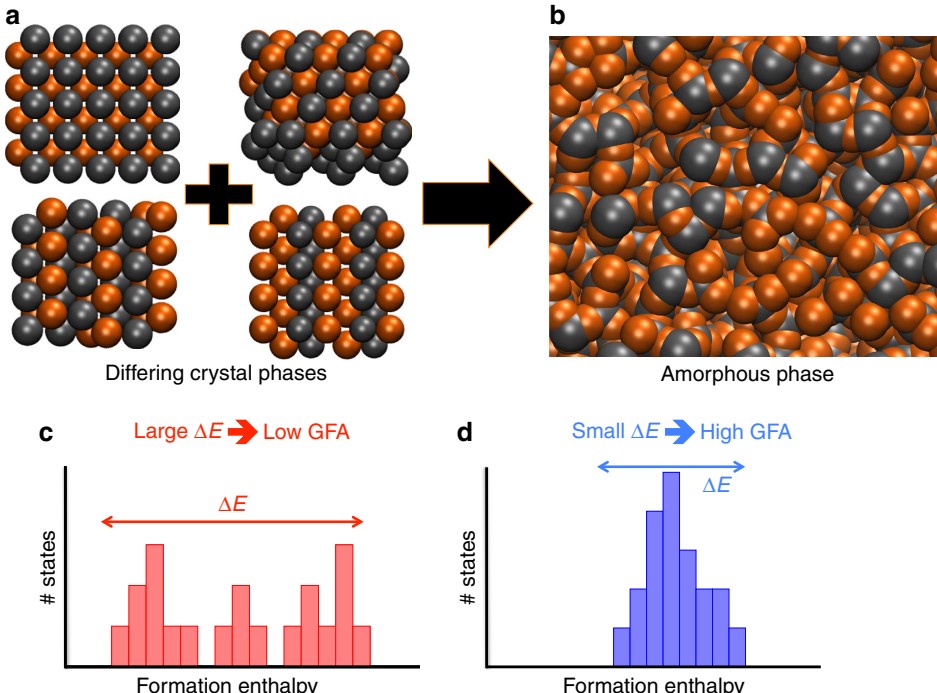

**Figure 1 | Descriptor for confusion.** If a particular alloy composition exhibits many structurally different stable and metastable crystal phases, which have similar energies, these phases will compete against each other during solidification, disrupting and frustrating the nucleation and crystallization processes, ultimately leading to an amorphous structure. (**a**) Distinct crystalline competing phases, which may compete and lead to (**b**) an amorphous structure. GFA should be (**c**) absent or (**d**) present, when the thermodynamic density of states is low or high, respectively.

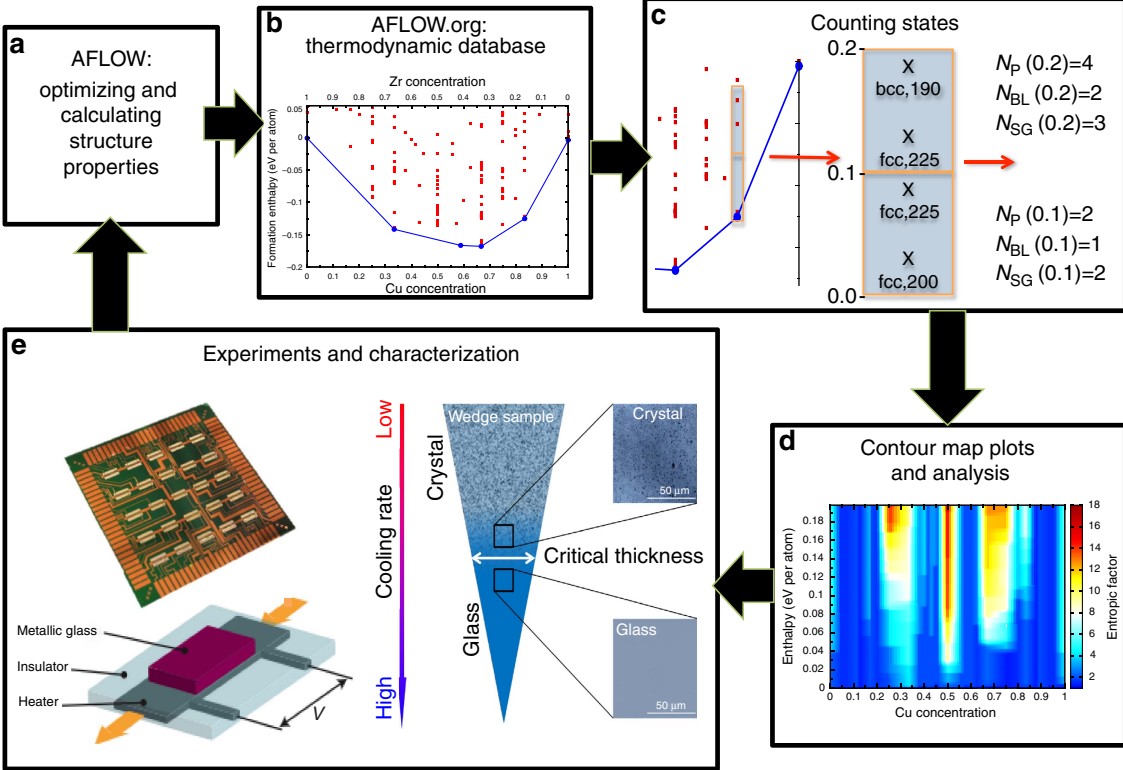

**Figure 2 | Integration of experimental and computational approaches.** (**a**) Multiple different structures for a given stoichiometry are built using the AFLOW prototype libraries[30], which are then optimized via VASP calculations under the AFLOW standard settings[53]. (**b**) The resulting data are added to the open thermodynamic database AFLOW[48,49]. (**c**) These data are accessed and used to obtain statistics on the cumulative distribution of entries ($N_P$), Bravais lattices types ($N_{BL}$) and space groups ($N_{SG}$) within a given formation enthalpy range (starting at zero). (**d**) Contour map plots are created from these distributions, allowing the identification of the best glass-forming alloys. (**e**) Finally, experimental synthesis and characterization are used to verify the computational results.

pure) inside a vacuum chamber with a base pressure better than $2 \times 10^{-7}$ Torr. Sputter deposition results in an effective quenching rate greater than $10^9$ K s$^{-1}$ (ref. 58), allowing a broad range of alloys to be obtained in the amorphous state.

Nanocalorimetry measurements were performed on thin-film samples of the binary alloys using micromachined calorimetry sensors[59–62]. The measurements were performed in vacuum at nominal heating rates ranging from 2,000 to 8,500 K s$^{-1}$, and cooling rates of ~5,000 K s$^{-1}$. All samples were repeatedly heated to 1,300 K to evaluate the crystallization behaviour both in the as-deposited state and after melt/quenching. Nanocalorimetry measurements reveal the glass transition, crystallization and liquidus temperatures. These quantities allow us to estimate GFA.

Figure 3a,b shows the nanocalorimetry results for the CuZr binary alloy with compositions in the bulk glass-forming region. Each measurement consisted of two thermal cycles in which the thin-film samples were heated to above the melting point and then quenched. All samples show clear signals corresponding to glass transition, crystallization and melting when first heated from the as-deposited state, indicating that they were deposited in the amorphous state (Fig. 3a). A better glass former has a lower critical cooling rate, so the amount of amorphous phase recovered after melt/quenching should scale with GFA. We observe in Fig. 3a that the magnitude of the crystallization peak after the first

thermal cycle changes significantly with composition: $Cu_{48.5}Zr_{51.5}$ has the strongest crystallization peak and is thus expected to have the highest GFA among the samples tested; $Cu_{55.5}Zr_{44.5}$ on the other hand has no discernible crystallization peak. This result is confirmed by calorimetry measurements obtained after cooling from the melted state: the heat released on solidification results in an exothermic peak in the cooling curve; the magnitude of this peak scales with the amount of crystalline phase formed on quenching and should be inversely proportional to the GFA (Fig. 3b). The experimentally observed number of phases and the amorphous phase thickness obtained from the casting experiments are shown in Fig. 3c. The calculated entropic factor, Fig. 3d, can be compared with these two quantities, and the results show very good agreement between all methods that $Cu_{50}Zr_{50}$ is the best glass-forming composition.

Figure 4 shows similar measurements for the NiZr alloy system, which has been shown to be a weak glass former[46,47]. Although as-deposited samples were amorphous and showed distinct crystallization peaks, subsequent melt/quenching did not produce any amorphous samples, and no crystallization peaks are observed in scans obtained after melting (Fig. 4a). Instead, we use $\gamma \equiv T_x/(T_g + T_l)$, defined in ref. 38 and shown in Fig. 4b, as a less direct measure of GFA. Figure 4b,c shows strong correlation between the experimental measurements and the entropic factor descriptor. There is a very weak

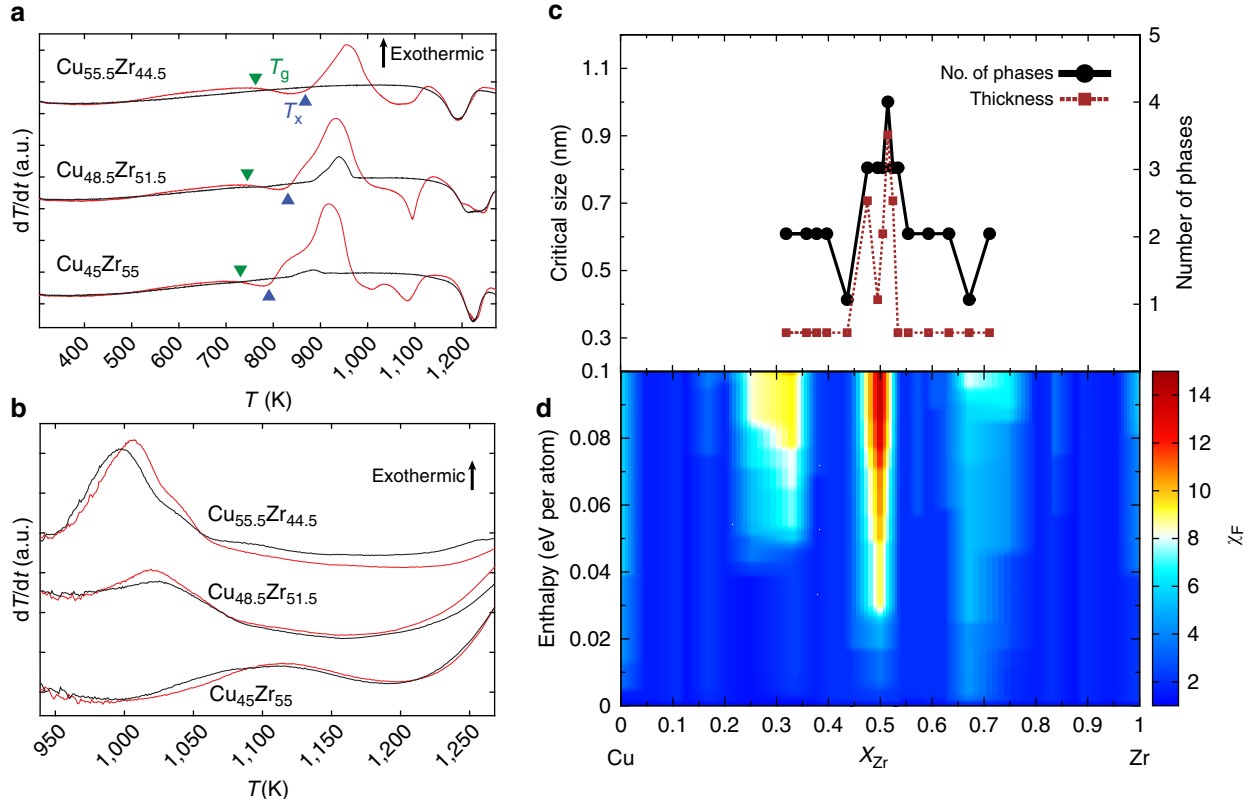

Figure 3 | Experimental and theoretical analysis of CuZr. (a) Nanocalorimetry measurements during heating and (b) cooling at different compositions. The first heating and cooling cycle measurements for the each composition are shown in red, and subsequent measurements are shown in black. (c) Number of phases (solid black line) as measured using X-ray diffraction, and thickness of the amorphous phase (dashed brown line), determined from the wedge shaped samples, as a function of composition. (d) Contour plot of the entropic factor as a function of formation enthalpy (zero corresponds to the ground state of the composition). The colour scale represents the entropic factor, calculated using equation (1), for each composition and formation enthalpy difference. This means that for a given fixed composition (x axis) all phases that are within a given formation enthalpy difference (y axis) from the ground state of that specific composition are used to compute the entropic factor (colour scale). Note the sharp peaks both in the number of states observed in experiment and in the entropic factor at the $Cu_{50}Zr_{50}$ composition, indicating that the descriptor correctly identifies this composition as having the highest GFA.

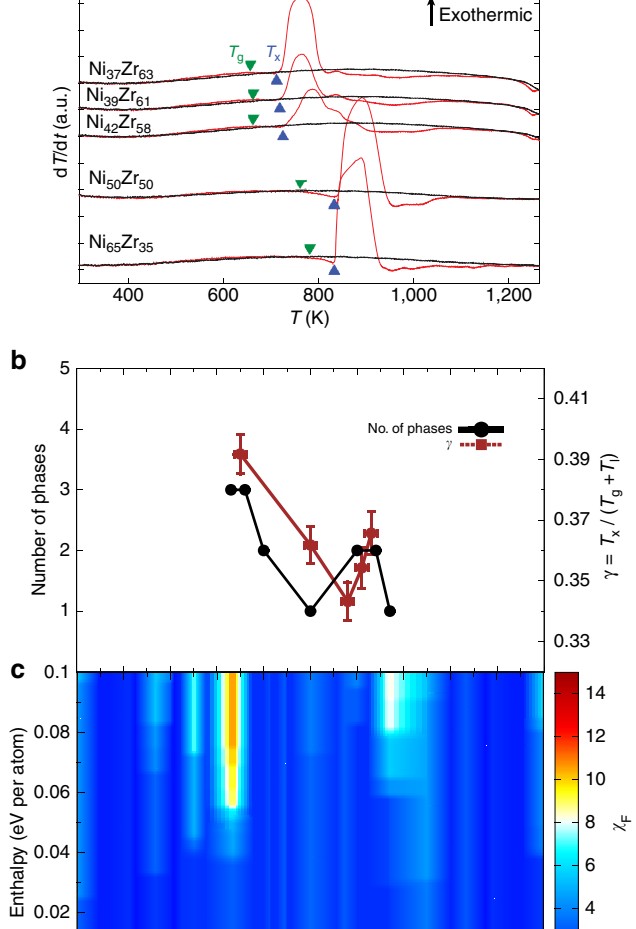

**Figure 4 | Experimental and theoretical analysis of NiZr.** (**a**) Nanocalorimetry measurements for NiZr during heating at different compositions. The first heating cycle measurements for each composition are shown in red, and subsequent measurements are shown in black. (**b**) Number of phases (solid black line) as measured using X-ray diffraction, and $\gamma$ descriptor calculated for NiZr alloys (dashed brown line). (**c**) Contour plot of the entropic factor as a function of formation enthalpy (zero corresponds to the ground state of the composition). Note the sharp peaks both in the number of states observed in experiment and in the entropic factor at the $Ni_{35}Zr_{65}$ and $Ni_{65}Zr_{35}$ compositions.

GFA peak around $Ni_{35}Zr_{65}$ according to the experimental measurements, which is predicted to be around $Ni_{30}Zr_{70}$ by the entropic factor. A more pronounced peak is measured around $Ni_{65}Zr_{35}$, which is also successfully predicted in the same region by the entropic factor descriptor. Thus, the new proposed descriptor correlates well with the traditional empirical indicators of glass formation in metallic alloys, with an accuracy of the order of 5% in composition, which is quite satisfactory. In addition, comparing Figs 3c and 4c, it is clear that the entropic factor exhibited by the high GFA alloy CuZr is significantly higher than that shown by the low GFA NiZr alloy, thus correctly pointing out the more favourable alloy system for glass formation. These results validate our ansatz and show that crystalline phase data can be used to predict the formation of amorphous phases.

**Descriptor for GFA.** Following this demonstration of the promise of our characterization of structural confusion, we proceed to enhance it into a broader and more quantitative model. This requires several steps: the ansatz is that the presence of highly dissimilar structures with very similar enthalpy correlates with GFA and the descriptor should contain factors describing enthalpy proximity, structural similarity and appropriate normalizations. Once the descriptor is defined, it will be confronted with experimental results and a threshold will be found self consistently. Finally, the formalism will be applied to our online repository AFLOW for appropriate statistical analysis and potential suggestions of glass-forming alloys.

**Enthalpy proximity.** The descriptor should favour states with enthalpy close to the ground state. This is captured by a Boltzmann factor:

$$
\begin{aligned}
f(H_i) \;=\; & \exp\left(\frac{-|H_i - H_0|}{k_B T_0}\right) \times \\
& \times \begin{cases} 1, & H_i < 0 \\ e^{-H_i/k_B T_0}, & 0 \le H_i < 50\,\text{meV} \\ 0, & 50\,\text{meV} \le H_i \end{cases}
\end{aligned} \quad (2)
$$

in which $H_0$ is the lowest enthalpy for a given concentration, and $T_0$ is room temperature. The inclusion of phases with positive formation enthalpy is necessary due to glass formation occurring at higher temperatures, at which higher enthalpy phases become accessible[63]. The cutoff value for including positive formation enthalpy phases is taken to be 50 meV $\sim 600\,\text{K}$, of the same order as the glass transition temperature of several metallic glasses.

**Structure similarity.** To correlate properties of structures having different decorations of the underlying lattice, we use a lattice-free formalism, the expansion in local atomic environments (AEs)[64]. The AE of an atom is defined as the polyhedron formed by the atoms present in the neighbourhood up to the distance of the maximum gap in the radial distribution function. A given structure has the corresponding AE calculated for each and every unique atom and then is expanded as:

$$
|\psi\rangle = \sum c_i |AE_i\rangle \quad \langle AE_i|AE_j\rangle = \delta_{ij},
$$
$$
c_i = \langle AE_i|\psi\rangle, \quad \sum c_i^2 = 1 \quad (3)
$$

where $\psi$ is a vector representing a given atomic structure. In this representation, the scalar product

$$
\langle\psi|\psi'\rangle = \sum_{ij}\left\langle AE_i \middle| c_i^* c_j' \middle| AE_j \right\rangle = \sum_i c_i^* c_i', \quad (4)
$$

is used to quantify the structural (dis)similarity between two distinct structures. The structural similarity factor is taken as an exponential having the maximum when $\langle\psi_i|\psi_0\rangle = 0$ (structures are dissimilar) and decaying to 0 at $\langle\psi_i|\psi_0\rangle = 1$ (structures are similar):

$$
\begin{aligned}
g(|\psi_i\rangle) \;=\; & \exp\left(\frac{-\theta}{|1-\langle\psi_i|\psi_0\rangle|} + \theta|1-\langle\psi_i|\psi_0\rangle|\right) \times \\
& \times \left(1 - \overline{\langle\psi_i|\psi_j\rangle}\right)^2,
\end{aligned} \quad (5)
$$

where $\theta = 0.25$ is a constant, based on an analysis of the available experimental data and kept constant for the entire study. The multiplicative coefficient is added to take into account the limitation that the exponential is taken with respect to the lowest enthalpy state at a given concentration $\psi_0$, and therefore structural similarity among metastable states needs to be accounted for by taking the average scalar product between metastable structures $i$ and $j$, $\overline{\langle\psi_i|\psi_j\rangle}$, computed over all possible combinations for a given stoichiometry $\{x\}$.

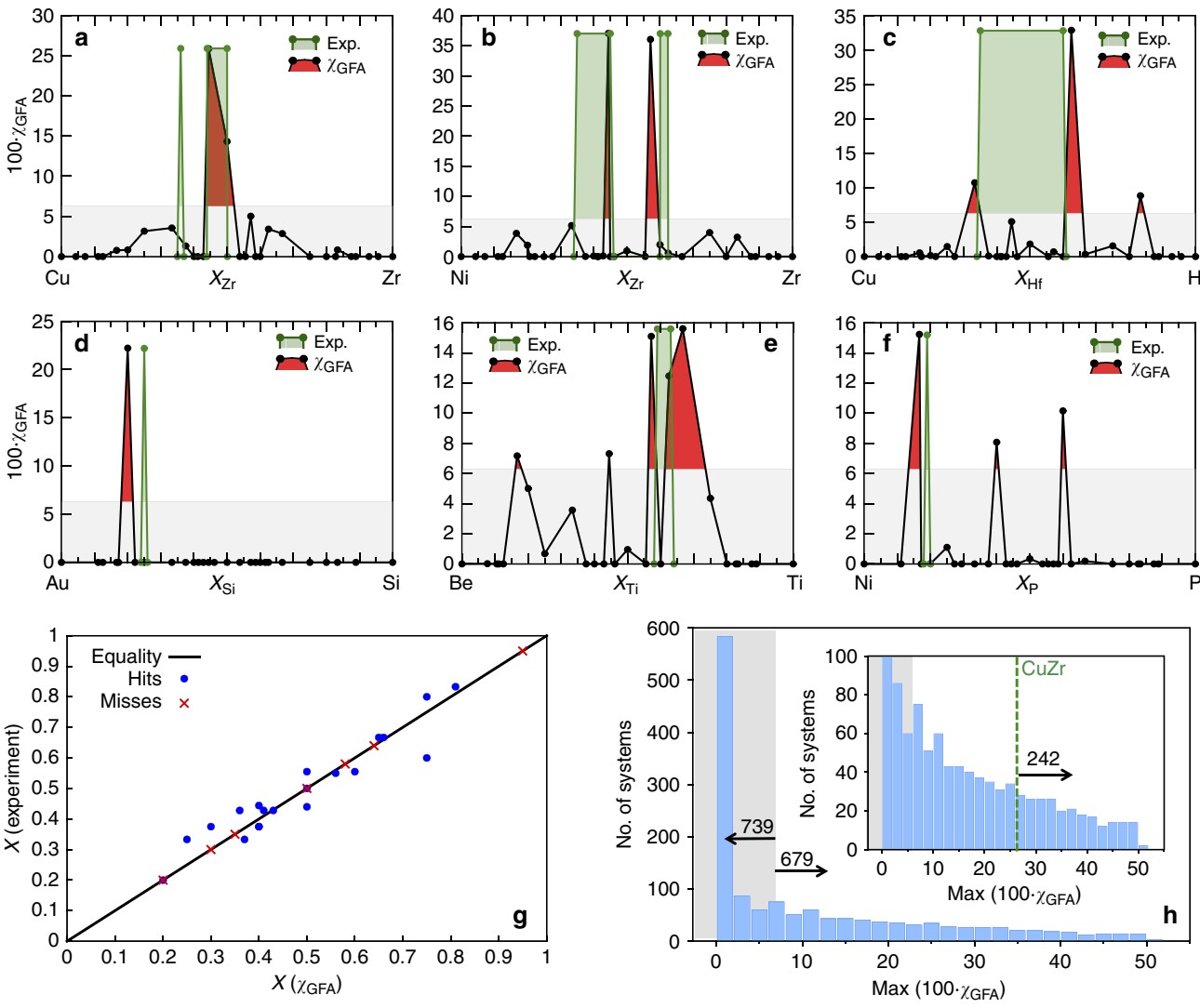

**Figure 5 | GFA descriptor spectra for different alloys.** Predictions are shown in black line/solid red fill, experimentally reported compositions are shown in green line/transparent green fill and the area under the threshold is shown in grey. (**a**) CuZr (reported glass formers Cu$_{50}$Zr$_{50}$, Cu$_{56}$Zr$_{44}$ and Cu$_{64}$Zr$_{36}$ (refs 42,79)); (**b**) NiZr (reported glass formers Ni$_{1-x}$Zr$_x$ with 0.35 < $x$ < 0.45 and 0.60 < $x$ < 0.63 (ref. 46)); (**c**) CuHf (reported glass former Cu$_{1-x}$Hf$_x$ with 0.35 < $x$ < 0.60 (ref. 65)); (**d**) AuSi (reported glass former Au$_{75}$Si$_{25}$ (ref. 66)); (**e**) BeTi (reported glass former Be$_{1-x}$Ti$_x$ with 0.59 < $x$ < 0.63 (ref. 67)); (**f**) NiP (reported glass former Ni$_{81}$P$_{19}$ (ref. 68)). (**g**) Reported versus predicted glass-forming concentrations for the 16 training systems. Missed glass formers are noted as red crosses. (**h**) Statistical distribution of the maximum peak GFA value for 1,418 different binary alloys. Inset shows a close up of the same plot. Area under the threshold is shown in grey.

**Normalization.** The normalization is represented by this expression computed for each stoichiometry $\{x\}$ of a given alloy system:

$$h(\{x\}) = \text{No. of entries within cutoff at stoich. } \{x\} \quad (6)$$

**GFA descriptor.** Combining equations (2), (5) and (6) we generate the GFA descriptor evaluated by summing through structures $i$ at a fixed stoichiometry $\{x\}$:

$$\chi_{\text{GFA}}(\{x\}) = \frac{\sum_i f(H_i) g(|\psi_i|)}{h(\{x\})}. \quad (7)$$

A large peak of $\chi_{\text{GFA}}(\{x\})$ is expected to indicate good GFA at a particular concentration $\{x\}$.

**Comparison with experiments and threshold value.** The GFA descriptor $\chi_{\text{GFA}}(\{x\})$ was trained with respect to the available

experimental data on binary metallic glasses. These data are scarce and sparse. Usually, only glass-forming compositions are reported[42,46,65–78], hindering the training of the descriptor to determine true negatives. Equipped with these 16 systems' comparisons, we search for a threshold, which is found self-consistently as the lowest value maximizing the ratio 'peak hits versus misses' without increasing the number of false positives. The threshold is found to be ~0.063. Figure 5a–f shows six binary examples comparing the predicted glass-forming compositions versus known experimental ones (arbitrarily assigned the highest descriptor value obtained for each corresponding system). The systems are CuZr (refs 42–45,79), NiZr (ref. 46), CuHf (refs 65,80,81), AuSi (ref. 66), BeTi (ref. 67) and NiP (ref. 68). When $\chi_{\text{GFA}} > 0.063$ we claim the existence of a glassy phase. As mentioned earlier, CuZr is probably the most studied binary metallic glass, due to its high GFA and accessible constituent materials. Figure 5a compares our prediction with the experimentally reported glass-forming compositions of Cu$_{50}$Zr$_{50}$,

$Cu_{56}Zr_{44}$ and $Cu_{64}Zr_{36}$ (refs 42–45,79), showing good agreement. For the CuHf alloy system, a glass-forming range is reported in $Cu_{1-x}Hf_x$ between $0.35 < x < 0.60$ (refs 65,80,81). As shown in Fig. 5c, we only register peaks at the extremes of this range, possibly suggesting a two glass coexistence in that composition range. Overall, of the 16 systems we analyse, 15 are correctly identified as glass formers with our descriptor (reliability $\sim$94%). However, not all of the peaks are reproduced. Out of the 26 peaks available in the 16 systems, 19 are found (reliability $\sim$73%). Qualitatively, the predicted concentrations are always close to the experimental values but due to the finite set of compositions spanned and the limited number of structures at each composition in our AFLOW repository, they are not strictly accurate. Figure 5g shows the correlation between predicted and

reported concentrations, which is quite good, with a root mean squared deviation of 5.4% for the successfully predicted ones (the AFLOW database has 200–250 different optimized structures for each of these systems. Several concentrations are computationally challenging to parameterize, hindering a uniform sampling of the spectrum.). Table 1 lists the systems and compositions used for the development of $\chi_{GFA}$.

## Discussion

The AFLOW repository, containing a total of 1,418 binary systems characterized by more then $330{,}000+$ appropriate structural entries, was screened using the new descriptor. The calculated $\chi_{GFA}(\{x\})$ spectra for all of the binaries are summarized in Fig. 5h. *In brevis*, the histogram of the maxima of $\chi_{GFA}$ shows that most of the systems, $\sim$52% (739 out of 1,418), are below the threshold and therefore expected to be non-glass formers. However, there are still many, $\sim$48% (679), above the threshold and therefore potential glass-forming systems. In particular $\sim$17% (242) have $\max(\chi_{GFA})$ higher than the known good glass-former CuZr, and hence are highly plausible candidates for metallic glass formers. The magnitude and sharpness of $\chi_{GFA}$ lead us to suggest the systems listed in Table 2 for further experimental validation. The predicted GFA spectra for the suggested systems can be found in Supplementary Figs 1 and 2.

Overall, our analysis implies that the existence of metallic glass phases could be a very common phenomenon in nature and that the missed experimental observations would be mostly due to the difficulty in achieving the appropriate quenching rates and/or in the choice of compositions. For addressing the latter problem, the rational interrogation of online repositories through carefully trained heuristic descriptors that capture the physical essence of the problem could become the long sought quantum leap in the field.

We propose a novel predictor for metallic glass formation that is based on the structural and thermodynamic properties of

### Table 1 | Experimentally reported glass formers.

| Reported | Predicted | References |
|---|---|---|
| $Cu_{50}Zr_{50}$, $Cu_{56}Zr_{44}$ and $Cu_{64}Zr_{36}$ | $Cu_{100-n}Zr_n$, $50 < n < 55$ | 42,79 |
| $Ni_{100-n}Zr_n$ $35 < n < 45$, $60 < n < 63$ | $Ni_{42.8}Zr_{57.2}$ $Ni_{55.5}Zr_{44.5}$ | 46 |
| $Cu_{100-n}Hf_n$ $35 < n < 60$ | $Cu_{16.7}Hf_{83.3}$ $Cu_{37.5}Hf_{62.5}$ $Cu_{66.7}Hf_{33.3}$ | 65 |
| $Au_{75}Si_{25}$ | $Au_{80}Si_{20}$ | 66 |
| $Be_{100-n}Ti_n$ $59 < n < 63$ | $Be_{33.3}Ti_{66.7}$ $Be_{42.8}Ti_{57.2}$ | 67 |
| $Ni_{81}P_{19}$ | $Ni_{60}P_{40}$ $Ni_{40}P_{60}$ $Ni_{83.3}P_{16.7}$ | 68 |
| $Au_{20}La_{80}$ | $Au_{20}La_{80}$ $Au_{37.5}La_{62.5}$ $Au_{62.5}La_{37.5}$ $Au_{80}La_{20}$ | 69 |
| $Au_{35}Ni_{65}$ | — | 70 |
| $Be_{100-n}Zr_n$ $50 < n < 70$ | $Be_{37.5}Zr_{62.5}$ | 67 |
| $Cu_{50}Ti_{50}$, $Cu_{58}Ti_{42}$, $Cu_{66}Ti_{34}$ | $Cu_{37.5}Ti_{62.5}$, $Cu_{66.7}Ti_{33.3}$ | 71 |
| $Nb_{30}Ni_{70}$, $Nb_{40.5}Ni_{59.5}$ | $Nb_{44.4}Ni_{55.6}$ $Nb_{50}Ni_{50}$ $Nb_{62.5}Ni_{37.5}$ $Nb_{83}Ni_{17}$ | 72,73 |
| $Ni_{60}Ta_{40}$ | $Ni_{55.6}Ta_{44.4}$ | 74 |
| $Ni_{40}Ti_{60}$ | $Ni_{16.7}Ti_{83.3}$ $Ni_{25}Ti_{75}$ $Ni_{37.5}Ti_{62.5}$ $Ni_{50}Ti_{50}$ $Ni_{55}Ti_{45}$ $Ni_{66.7}Ti_{33.3}$ | 75 |
| $Pd_{100-n}Si_n$ $5 < n < 25$ | $Pd_{60}Si_{40}$ | 76 |
| $P_{25}Pt_{75}$ | $P_{20}Pt_{80}$ $P_{33.3}Pt_{66.7}$ $P_{44}Pt_{56}$ | 77 |
| $Fe_{100-n}Zr_n$ $57 < n < 80$ | $Fe_{42.8}Zr_{57.2}$ | 78 |

List of 16 reported glass-forming alloys used for training the spectral descriptor. Whenever a broad glass-forming region was reported we counted two peaks, one at the beginning of the region and one at the end. This approach leads to a total of 26 peaks used as references. An empty entry at the second column means that no glass-forming composition was predicted, that is, a miss. The second column includes both peaks that correspond to the reported ones, as well as a few that do not correspond to any of the reported glass-forming alloys.

### Table 2 | Potential candidate glass formers.

| Glass forming compositions |
|---|
| $Al_{37.5}La_{62.5}$ |
| $Al_{60}Re_{40}$ |
| $As_{44.4}Nb_{55.6}$; $As_{60}Nb_{40}$ |
| $Co_{33}Zn_{67}$ |
| $As_{20}Pd_{80}$; $As_{62.5}Pd_{37.5}$ |
| $Ba_{83.3}Zn_{16.7}$ |
| $Be_{55}V_{45}$ |
| $Bi_{60}Pt_{40}$ |
| $Cr_{44.4}Rh_{55.6}$ |
| $Fe_{37.5}Nb_{62.5}$ |
| $Fe_{40}P_{60}$; $Fe_{62.5}P_{37.5}$ |
| $Ga_{40}Ir_{60}$ |
| $Ge_{62.5}Rh_{37.5}$ |
| $Hf_{44.4}Pd_{55.6}$ |
| $Hf_{55.5}Re_{44.5}$; $Hf_{60}Re_{40}$ |
| $La_{60}Pb_{40}$ |
| $La_{60}Pd_{40}$ |
| $Mg_{40}Pb_{60}$ |
| $Mn_{62.5}Si_{37.5}$ |
| $Nb_{55.5}Os_{44.5}$ |
| $Nb_{37.5}Si_{62.5}$ |
| $P_{83.3}Pd_{16.7}$ |
| $Pb_{62.5}Sc_{37.5}$; $Pb_{80}Sc_{20}$ |
| $Pd_{44.4}Zn_{55.6}$; $Pd_{60}Zn_{40}$ |
| $Pd_{37.5}Zr_{62.5}$; $Pd_{55.5}Zr_{44.5}$ |

List of unreported compositions that are predicted to present high GFA (spectra are shown in Supplementary Figs 1 and 2).

competing crystalline phases, which we calculate from first principles. This predictor stems from the concept that competition between energetically similar crystalline phases frustrates crystallization and thus promotes glass formation. It was developed into a robust numerical descriptor using formation enthalpies and structural similarity measures based on atomic environments. Detailed nanocalorimetry experiments verify the validity of this approach for two model systems, CuZr and NiZr. The non-reliance on experimental data allows for the construction of GFA spectra for 1,418 different binary alloy systems, by leveraging extensive libraries of computed crystalline phase data such as AFLOW. Our results predict that 17% of binary alloy systems are capable of glassifying, including many whose synthesis has not been previously reported in the literature, suggesting that there is great uncharted potential for new discoveries in this field.

## Methods

**Sample preparation.** The ingots of CuZr and NiZr alloys were prepared by arc-melting the pure elements under an argon atmosphere. The alloys were re-melted and suction cast into a wedge-shaped cavity in a copper mould. The as-cast rods were cut into half along the longitudinal direction and polished to a mirror finish followed by etching.

We also synthesized thin-film samples deposited by magnetron-sputtering elementary targets (99.99% pure) inside a vacuum chamber with a base pressure better than $2 \times 10^{-7}$ Torr. Sputter deposition results in an effective quenching rate $> 10^9 \, \mathrm{K \, s^{-1}}$ (ref. 58), allowing a broad range of alloys to be obtained in the amorphous state.

**Nanocalorimetry experiments.** Nanocalorimetry measurements were performed on thin-film samples of the binary alloys using micromachined calorimetry sensors[59–62]. The measurements were performed in vacuum at nominal heating rates ranging from 2,000 to 8,500 K s$^{-1}$, and cooling rates of $\sim 5,000$ K s$^{-1}$. All samples were repeatedly heated to 1,300 K to evaluate the crystallization behaviour both in the as-deposited state and after melt/quenching.

**First-principle calculations.** All density functional theory calculations were carried in accordance with the AFLOW standard settings, which are described in detail in ref. 53.

**Calculation of AEs.** To discern AEs we generate $N \times N \times N$ supercells for the structures under consideration, $N$ being odd and larger than or equal to 3. All distances are calculated with respect to the atoms in the central cell and only within a sphere centred on each atom with a radius chosen so as to guarantee that it is always enclosed by the supercell. If <100 neighbours are contained within this sphere then the supercell size is increased to meet this requirement. This is done to guarantee sufficient sampling, as well as to avoid spurious gaps around the edges of the supercell. Exceptions to this rule are considered when either there are two or more gaps of similar size, or when the AE defined by this rule generates convex polyhedra in which atoms are contained on the faces (instead of exclusively in the vertices). For the first case, two gaps are considered equivalent if they differ by 0.05 Å or less. In this case we adopt the gap which defines the smaller AE[82]. For the second case, whenever atoms are detected within a surface, the AE is reconstructed using the largest gap which defines an AE smaller than the initial one. After generating an AE, each of its vertices (atoms) are classified by the number and type of different faces (either triangular or quadrilateral) meeting at that point. Finally, an AE is described in terms of the number of each type of vertex[83]. It should be emphasized that, using this classification, slight distortions on the AEs are completely ignored, and thus we account only for significant differences in crystal structures.

**Data availability.** All the *ab initio* alloy data are freely available to the public as part of the AFLOW online repository and can be accessed through www.aflow.org following the REST-API interface[48].

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

## Acknowledgements

This work was supported by the National Science Foundation under DMREF Grants No. DMR-1436151, DMR-1436268 and DMR-1435820. Experiments were performed in part at the Center for Nanoscale Systems at Harvard University (supported by NSF under Award No. ECS-0335765), at the Materials Research Science and Engineering Center at Harvard University (supported by NSF under Award No. DMR-1420570) and at the Materials Research Science and Engineering Center at Yale University (supported by NSF under Award No. DMR-1119826). Calculations were performed at the Duke University—Center for Materials Genomics. E.P., C.T. and S.C. acknowledge partial support by the DOD-ONR (N00014-13-1-0635 and N00014-14-1-0526).

## Author contributions

S.C. proposed the entropy and spectral descriptors; E.P. wrote the code under the supervision of C.T. and O.L.; D.L. performed the nanocalorimetry experiments under the supervision of J.V.; Y.L., P.G. and Y.L. executed the combinatorial synthesis and X-ray diffraction and scanning electron microscopy measurements under the supervision of J.S. All authors contributed to the discussion and to the writing of the manuscript.

## Additional information

**Competing financial interests:** The authors declare no competing financial interests.

