## [Peer Review file · Nature Communications]

Reviewers' Comments:

Reviewer #1 (Remarks to the Author)

This paper attempts to use an "entropic factor" to interpret the "confusion principle". The new factor basically counts the number of phases, types of lattice and space group that may be competing with the glass. A high value is believed to confuse the crystallization and favor GFA. The competitors were determined by their enthalpy-degeneracy, assessed using ab initio database.

1) This is a useful attempt. It adds a few metastable structures as potential competing phases, from the enthalpy standpoint.

2) But with regards to predicting GFA, it is a "modification", rather than a breakthrough. For example, even with just the equilibrium phases seen in the phase diagram, or the Trg (reduced glass transition temperature) that reflects how well the liquid would be competing with crystallizing phases, the prediction of GFA is already pretty good. See WL Johnson's Nat Commun paper this year. Also, in the current manuscript the experimental "# of phases" peak already matches the one from experiments and from "entropic factor". So it is not clear that the ab initio candidates actually added something essential and helped much in correlating with GFA.

3) The confusion cannot just be due to "too many candidates with similar enthalpy". For example, if we have quite a number of enthalpy-degenerate candidates but one of them is particularly advantageous in nucleation not due to enthalpy but because it has a low interface energy or fast nucleation kinetics, or ..., then it breaks the confusion and kills GFA. The more candidates, the more likely that this could happen. For GFA, this is worse than the case of very few candidates that are known to nucleate very slowly. Perhaps a potential quasicrystal with low nucleation barrier due to low interface energy is such an example.

4) In general, the confusion can be due to kinetics, not necessarily how crowded near the ground state. For example, the compositional partitioning needed can be sluggish, even if you only have two phases (but with quite different compositions) to nucleate into.

5) The authors only did two binary systems, CuZr and NiZr. It is not clear that this method has merit for general glass-forming systems. It is limited to finding a peak within a system, relative to neighboring compositions. How do we compare different systems? Apparently two systems with similar "thermodynamic density" can have very different GFA.

6) Some groups emphasize the amorphous side (such as its structure, liquid viscosity-fragility,...), and this time this paper focuses on the crystalline state. I am however of the opinion that both should be in the picture, for GFA.

In summary, I feel that this work is of preliminary nature and the conclusion is a bit too simplified. The predictive power mentioned in the title seems to be an overclaim for Nature Commun. The result is more like an incremental step in expanding what should be considered when dealing with glass-crystal competition.

Reviewer #2 (Remarks to the Author)

The formation of metallic glasses still remains mostly unclear. This lack of knowledge hinders the exploration for new systems, still performed with combinatorial trial and error. This article propose a heuristic descriptor quantifying such issue based on the \thermodynamic density of competing crystalline states, parameterized from high-throughput ab-initio calculations. The experimental results corroborate the capability of the heuristic descriptor in predicting glass forming ability through the compositional space. The results is expected to deepen the understanding of the underlying mechanisms and to accelerate the discovery of novel metallic glasses. It is a good try using the high-throughput ab-initio calculations to explain and understand the metallic glass formation. I recommend the paper for acceptance of publication in Nature Communications.

Reviewer #3 (Remarks to the Author)

Report of Manuscript: "Predicting Bulk Metallic Glass Forming Ability with the Thermodynamic Density of Competing Crystalline States", by Eric Perim et al.

This paper reports the entropic factor describing the glass forming ability, using two Cu-Zr and Ni-Zr systems to test its validity. This entropic factor is based on previous proposal "confusion" idea. This work demonstrates that it is very hard task to get the entropic factor for a particular alloy composition, which hinders its wide application to predict a new glass former alloy. Furthermore, only two alloys, and small composition ranges shown in Figs. 3 and 4 studied in this manuscript, are far enough to validate the agreement of glass forming ability with the entropy factor suggested in this work. All-in-all, I am of the opinion that the manuscript does not meet the standards of Nature Communications.

REVIEWERS' COMMENTS:

Reviewer #1 (Remarks to the Author):

The authors have addressed the concerns I raised in the first round of review. The data set has been expanded to a large number of systems and the descriptors have been further developed and explained. This manuscript is now acceptable for publication in Nat Commun.

Reviewer #3 (Remarks to the Author):

This revised manuscript addressed some questions which referees asked during the first run. However, the novelty of this manuscript is still missing although it provides some descriptors for bulk metallic glasses based on reported data. In fact, the authors predicted some possible good BMG systems using their descriptors. Thus, it will be nature for authors to fabricate such possible good BMG systems to confirm the novelty reported in this revised manuscript.

Reponses to the referees

Reviewer #1:

This paper attempts to use an "entropic factor" to interpret the "confusion principle". The new factor basically counts the number of phases, types of lattice and space group that may be competing with the glass. A high value is believed to confuse the crystallization and favor GFA. The competitors were determined by their enthalpy-degeneracy, assessed using ab initio database.

1) This is a useful attempt. It adds a few metastable structures as potential competing phases, from the enthalpy standpoint.

2) But with regards to predicting GFA, it is a "modification", rather than a breakthrough. For example, even with just the equilibrium phases seen in the phase diagram, or the Trg (reduced glass transition temperature) that reflects how well the liquid would be competing with crystallizing phases, the prediction of GFA is already pretty good. See WL Johnson's Nat Commun paper this year. Also, in the current manuscript the experimental "# of phases" peak already matches the one from experiments and from "entropic factor". So it is not clear that the ab initio candidates actually added something essential and helped much in correlating with GFA.

Authors

The reviewer points out a weakness of the manuscript in its first version: the limited use of *ab-initio* data. The main point here is that our descriptor solely uses *ab-initio* calculation results on crystalline phases to predict GFA, without any reliance on the measurement of complex experimental parameters, such as the T_g . It is therefore uniquely suitable for materials discovery and design purposes. The fact that it closely matches those previous empirical quantities used to describe GFA in known glass-forming systems is not a drawback but a necessary requirement to demonstrate that such a simple descriptor is capable of capturing the essential physical content of these complex empirical quantities, which are manifestly unsuitable for *a priori* identification of new metal-forming alloys. Moreover, the substantial expansion of our submission clearly presents the utility of our model for large scale screening of multiple potential glass-forming candidates, which is impractical by exclusive reliance on the current empirical GFA correlators.

Now, the current version of the manuscript contains a second descriptor, the spectral evolution of the "entropy factor" trying to characterize the capability of the alloy in forming glasses. This second descriptor is based on structural and enthalpic mismatches and it is trained with the available list of known binary glasses. It is then used to predict novel glasses from a well established quantum-mechanical data repository. The highly efficient and effective use of *ab-initio* data and this descriptor is described on pages 5-8.

Reviewer #1:

3) The confusion cannot just be due to "too many candidates with similar enthalpy". For example, if we have quite a number of enthalpy-degenerate candidates but one of them is particularly advantageous in nucleation not due to enthalpy but because it has a low interface energy or fast nucleation kinetics, or ..., then it breaks the confusion and kills GFA. The more candidates, the more likely that this could happen. For GFA, this is worse than the case of very few candidates that are known to nucleate very slowly. Perhaps a potential quasicrystal with low nucleation barrier due to low interface energy is such an example.

4) In general, the confusion can be due to kinetics, not necessarily how crowded near the ground state. For example, the compositional partitioning needed can be sluggish, even if you only have two phases (but with quite different compositions) to nucleate into.

6) Some groups emphasize the amorphous side (such as its structure, liquid viscosity-fragility,...), and this time this paper focuses on the crystalline state. I am however of the opinion that both should be in the picture, for GFA.

Authors

The reviewer correctly pinpoints possible other possible factors contributing to the GFA. We cannot rule them out, but they are out of the reach of a high-throughput quantum analysis leading to the identification of novel systems, by simple and quick *ab-initio* calculations. We believe that the predictive power, estimated in $\sim 75\%$, warrants occasional false positives. The crucial point here is not to exhaustively describe the glass-formation process, with all its complexities, but to present a model that captures enough of the essential physics and is yet sufficiently simple to be practically employed for computationally guided design of new BMG's. Despite the limits in our methodology, we believe that we demonstrate convincingly that it is interesting enough to deserve publication.

Reviewer #1:

5) *The authors only did two binary systems, CuZr and NiZr. It is not clear that this method has merit for general glass-forming systems. It is limited to finding a peak within a system, relative to neighboring compositions. How do we compare different systems? Apparently two systems with similar "thermodynamic density" can have very different GFA.*

In summary, I feel that this work is of preliminary nature and the conclusion is a bit too simplified. The predictive power mentioned in the title seems to be an overclaim for Nature Commun. The result is more like an incremental step in expanding what should be considered when dealing with glass-crystal competition.

Authors

We agree with the referee's comment on the preliminary flavor of the original submission and the limited number of systems studied in it. The current version is drastically expanded. Now we are confident that the current analysis of all possible binary systems (1400+ instead of two), with a spectral descriptor based on energetic and structural considerations, trained with ~ 20 experimental reports (Table I) and with quite a few potential novel glasses (Table II), would not be considered incremental. To the best of our knowledge, nobody has ever challenged the problem in a similar way. Some potential novel BMG candidates are presented in Table II of the revised manuscript:

Glass forming concentrations	
$\text{Al}_{37.5}\text{La}_{62.5}$	$\text{Ge}_{62.5}\text{Rh}_{37.5}$
$\text{Al}_{60}\text{Re}_{40}$	$\text{Hf}_{44.4}\text{Pd}_{55.6}$
$\text{As}_{44.4}\text{Nb}_{55.6}$; $\text{As}_{60}\text{Nb}_{40}$	$\text{Hf}_{55.5}\text{Re}_{44.5}$ - $\text{Hf}_{60}\text{Re}_{40}$
$\text{Co}_{33}\text{Zn}_{67}$	$\text{La}_{60}\text{Pb}_{40}$
$\text{As}_{20}\text{Pd}_{80}$; $\text{As}_{62.5}\text{Pd}_{37.5}$	$\text{La}_{60}\text{Pd}_{40}$
$\text{Ba}_{83.3}\text{Zn}_{16.7}$	$\text{Mg}_{40}\text{Pb}_{60}$
$\text{Be}_{55}\text{V}_{45}$	$\text{Mn}_{62.5}\text{Si}_{37.5}$
$\text{Bi}_{60}\text{Pt}_{40}$	$\text{Nb}_{55.5}\text{Os}_{44.5}$
$\text{Cr}_{44.4}\text{Rh}_{55.6}$	$\text{Nb}_{37.5}\text{Si}_{62.5}$
$\text{Fe}_{37.5}\text{Nb}_{62.5}$	$\text{P}_{83.3}\text{Pd}_{16.7}$
$\text{Fe}_{40}\text{P}_{60}$; $\text{Fe}_{62.5}\text{P}_{37.5}$	$\text{Pb}_{62.5}\text{Sc}_{37.5}$; $\text{Pb}_{80}\text{Sc}_{20}$
$\text{Ga}_{40}\text{Ir}_{60}$	$\text{Pd}_{44.4}\text{Zn}_{55.6}$; $\text{Pd}_{60}\text{Zn}_{40}$
	$\text{Pd}_{37.5}\text{Zr}_{62.5}$; $\text{Pd}_{55.5}\text{Zr}_{44.5}$

TABLE II: List of unreported concentrations that are predicted to present high glass forming ability (spectra are shown in the supplemental information).

Reviewer #2

The formation of metallic glasses still remains mostly unclear. This lack of knowledge hinders the exploration for new systems, still performed with combinatorial trial and error. This article propose a heuristic descriptor quantifying such issue based on the thermodynamic density of competing crystalline states, parameterized from high-throughput ab-initio calculations. The experimental results corroborate the capability of the heuristic descriptor in predicting glass forming ability through the compositional space. The results is expected to deepen the understanding of the underlying mechanisms and to accelerate the discovery of novel metallic glasses. It is a good try using the high-throughput ab-initio calculations to explain and understand the metallic glass formation. I recommend the paper for acceptance of publication in Nature Communications

Authors

We thank Reviewer #2 for the supportive report. We invite him/her to read the second version of the paper, which has been drastically enhanced.

Reviewer #3

This paper reports the entropic factor describing the glass forming ability, using two Cu-Zr and Ni-Zr systems to test its validity. This entropic factor is based on previous proposal "confusion" idea. This work demonstrates that it is very hard task to get the entropic factor for a particular alloy composition, which hinders its wide application to predict a new glass former alloy.

Authors

We agree that the task if very difficult and therefore it needs to be addressed within appropriate approximations. As mentioned in the answer to Reviewer #1 we have extended our approach to include more effective and extensive use of *ab-initio* data.

Reviewer #3

Furthermore, only two alloys, and small composition ranges shown in Figs. 3 and 4 studied in this manuscript, are far enough to validate the agreement of glass forming ability with the entropy factor suggested in this work. All-in-all, I am of the opinion that the manuscript does not meet the standards of Nature Communications.

Authors

As mentioned in our response to referee #1 the revised manuscript is radically extended to address this point, and now includes an introduction of a second descriptor and an analysis of many more systems. The descriptor's spectral decomposition leads to a better determination of the concentration space. The method is completely *ab-initio* and does not require any input from experiments (except for the self-consistent determination for a threshold).

In summary, 1) The spectral descriptor is trained with ~20 experimental reports of binary metallic glasses (see Fig5g, attached below). 2) The spectra compared with experimental concentrations show good agreement, Fig5(a-f). 3) The descriptor was applied to our ab-initio repository AFLOW, containing 330,000+ calculations of binary systems, thus allows the analysis of important features, such as the frequency of metallic glasses versus solid solutions or intermetallics (Fig5h). We therefore believe that the revised manuscript fits the requirements of Nature Communications in terms of originality, advancement, and general interest.

FIG. 5: GFA descriptor spectra for different alloys. Predictions are shown in black line/solid red fill, experimentally reported compositions are shown in green line/transparent green fill and the area under the threshold is shown in grey. (a) CuZr (reported glass formers $\text{Cu}_{50}\text{Zr}_{50}$, $\text{Cu}_{56}\text{Zr}_{44}$ and $\text{Cu}_{64}\text{Zr}_{36}$ [44, 64]); (b) NiZr (reported glass formers $\text{Ni}_{1-x}\text{Zr}_x$ with $0.35 < x < 0.45$ and $0.60 < x < 0.63$ [48]); (c) CuHf (reported glass former $\text{Cu}_{1-x}\text{Hf}_x$ with $0.35 < x < 0.60$ [65]); (d) AuSi (reported glass former $\text{Au}_{75}\text{Si}_{25}$ [66]); (e) BeTi (reported glass former $\text{Be}_{1-x}\text{Ti}_x$ with $0.59 < x < 0.63$ [67]); (f) NiP (reported glass former $\text{Ni}_{81}\text{P}_{19}$ [68]). (g) Reported versus predicted glass forming concentrations for the 16 training systems. Missed glass formers are noted as red crosses. (h) Statistical distribution of the maximum peak GFA value for 1418 different binary alloys. Inset shows a close up of the same plot. Area under the threshold is shown in grey.